# Exosomes in Bone Sarcomas: Key Players in Metastasis

**DOI:** 10.3390/cells9010241

**Published:** 2020-01-17

**Authors:** Mariona Chicón-Bosch, Oscar M. Tirado

**Affiliations:** 1Sarcoma Research Group, Oncobell Program, Bellvitge Biomedical Research Institute (IDIBELL), L’Hospitalet de Llobregat, 08908 Barcelona, Spain; 2CIBERONC, Carlos III Institute of Health (ISCIII), 28029 Madrid, Spain; 3Institut Català d’Oncologia (ICO), L’Hospitalet de Llobregat, 08908 Barcelona, Spain

**Keywords:** bone sarcoma, exosomes, metastasis, osteosarcoma, Ewing sarcoma, chondrosarcoma

## Abstract

Bone sarcomas are rare cancers which often present with metastatic disease and are still associated with poor survival rates. Studies in the last decade have identified that exosomes, a type of extracellular vesicle released by cells, play an important role in tumour progression and dissemination. Through the transfer of their cargo (RNAs, proteins, and lipids) across cells, they are involved in cellular cross-talk and can induce changes in cellular behaviour. Exosomes have been shown to be important in metastasis organotropism, induction of angiogenesis and vascular permeability, the education of cells towards a pro-metastatic phenotype or the interaction between stromal and tumour cells. Due to the importance exosomes have in disease progression and the high incidence of metastasis in bone sarcomas, recent studies have evaluated the implications of these extracellular vesicles in bone sarcomas. In this review, we discuss the studies that evaluate the role of exosomes in osteosarcoma, Ewing sarcoma, and preliminary data on chondrosarcoma.

## 1. Bone Sarcomas

More than 50 different types of sarcomas have been identified, emerging in different tissues, with different genetic backgrounds, and at different ages [1]. Moreover, adding to this diversity, they are far less common than carcinomas. Although the introduction of chemotherapy has improved survival rates, treatment has not advanced in the last decades, and metastasis is still an important burden for patient survival [1]. It is estimated that one third of patients will not survive sarcomas due to metastatic spread. Amongst sarcomas, a subset arises in the bone tissue, named bone sarcomas. These are rare, with an incidence of 1–2 new cases per 100.000 per year and only account for <0.2% of registered tumours in EUROCARE [2,3]. The most frequent bone sarcomas are osteosarcoma (OS), chondrosarcoma (CS), and Ewing sarcoma (ES). Although some inherited and acquired genetic factors have been associated with the origin of these neoplasms, many do not have a cause yet, complicating their diagnosis and accurate treatment [4]. Therefore, there is an urgent need to find what is driving these bone sarcoma’s tumorigenesis and how to better inhibit and eliminate metastatic disease. Amongst the different factors that are being studied in tumour progression across different cancers, we find the transfer of genetic material through exosomes that cells release and take up. In this review, the role of exosomes in metastasis, with a focus on bone sarcomas, will be discussed.

## 2. Exosomes

Exosomes are a type of extracellular vesicle secreted through endosomes and play an important role in cellular cross-talk via the transfer of their cargo across cells [5]. They were first identified by Johnstone et al. [6], although at that time their function was associated with waste removal from cells. Since then, extensive research postulates exosomes as being important players in cellular communication between neighbouring cells as well as across distant sites [7]. Exosomes are characterised by having a size of 30–150 nm and a cup-shape appearance under an electron microscope [8]. Furthermore, a lipid bilayer membrane and an enrichment in small RNAs as part of their cargo, as well as different proteins, lipids, and even DNAs, distinguish exosomes from other extracellular vesicle types [5,9].

### 2.1. Exosome Biogenesis, Release, and Uptake

In contrast to microvesicles which are released from cells through outward budding of the plasma membrane [10], exosomes are released through an active process via endosomes [11]. In late endosomes, the inward budding of the endosomal membrane results in the accumulation of small vesicles transforming the endosome into a multivesicular body (Figure 1). At this stage, the specific sorting of lipids, proteins, and nucleic acids into these vesicles occurs. Two different mechanisms have been identified for this active sorting of the cargo: the endosomal sorting complex required for transport (ESCRT) [12] and the ESCRT-independent process, which can involve syndecans [13] or ceramides [14]. The involvement of different proteins and cellular machinery in the sorting of cargo into exosomes has been linked to different exosome subtypes, although this is still under investigation [15]. Moreover, although an important part of the cargo reflects the cell of origin, exosomes are not just identical mirrors of the cells from which they are released. The active sorting of the cargo results in a specific loading of certain RNAs and proteins that will differ from the cell of origin [16].

Once these small vesicles (now exosomes) are filled with different biological molecules, the multivesicular body that contains them will fuse with the plasma membrane, thus releasing the exosomes into the extracellular space (Figure 1) [10]. Similar to the sorting of the cargo, different cellular machineries have been associated with this process of exosome release [9]. Rab GTPases are the main family of proteins associated with this stage [17]. Different members have been identified to regulate the transport of multivesicular bodies and the fusion with the plasma membrane, and the involvement of different Rab proteins results in different protein profiles on the exosome membrane [18]. Therefore, there are different stages in the exosomes biogenesis and release that can influence the cargo and membrane composition of exosomes, resulting in a heterogeneous exosome population, partly resembling the cell of origin but still with some differences in the profile. 

Considering that exosomes are important factors in the cellular cross-talk, they need to go from producing cells (those that produce and release exosomes into circulation) to recipient cells. In general terms, exosomes can interact with recipient cells via three mechanisms: i) by internalization of their cargo via fusion, ii) via endocytosis, or iii) through interaction between the exosome and cellular membrane components (Figure 1). Therefore, the exosome cargo can be directly released into the cytoplasm of the recipient cell by the fusion of exosomes with the plasma membrane. In this line, a low pH in the recipient cell has been shown to facilitate this process [19]. The other mechanism by which exosomes transfer their cargo to recipient cells is by endocytosis, through which these small extracellular vesicles enter the cells [7]. Although it is still not clear how exosomes recognise the cells they need to interact with or whether this depends on exosome or cell surface markers [20], different cellular mechanisms have been associated with the internalization of exosomes, such as clathrin-dependent- [21] or caveolin-dependent [22]-mediated endocytosis. The last mechanism that has been associated with the intercellular communication between cells via exosomes is the interaction of surface-receptor proteins (Figure 1). As exosomes carry some lipids and proteins that resemble the cell of origin, such as MHC or oncoproteins, the interaction of these exosome membrane proteins with the receptors present in the plasma membrane of recipient cells could trigger a response resulting in cellular changes [23,24]. Therefore, via these processes, exosomes can interact with recipient cells, releasing the cargo or inducing signalling cascades that could ultimately result in changes in the cellular behaviour.

### 2.2. Exosome Cargo and Composition

The exosome composition can depend on the biogenesis, the environmental conditions, the physiological status, or the cell of origin. Nevertheless, there are some shared components that help define what exosomes are. As mentioned before, several proteins and cellular machinery are important in the biogenesis and release of exosomes, and in turn, become part of the exosome cargo or even exosome membrane. Amongst them, we find ESCRT-associated proteins, such as TSG-101 or Alix [25], members of the Rab GTPase family such as Rab7 or Rab11B [18], or tetraspanins, important in the formation of multivesicular bodies, including CD9, CD63, and CD81 [26]. Moreover, as they are produced through double inward budding, proteins contained in the exosome membrane come from the outer cellular membrane [7]. Therefore, they can be distinguished from other extracellular vesicles as they lack proteins characteristic of different cellular compartments, such as the Golgi or endoplasmic reticulum [27].

Exosomes are characterised by enrichment of small RNAs [9]. However, other types of genetic material have been identified in these vesicles. The first evidence that functional mRNAs were contained in exosomes was in 2007 by Valadi et al [28]. Since then, a wide variety of RNA species, including mRNA, miRNA, tRNA, and lncRNA have been identified in exosomes [5]. In addition, although less studied, DNA has been identified as part of the exosome cargo [29], even reflecting the mutational burden in oncogenes and tumour suppressor genes in exosomes from cancer samples [30].

All this cargo is encapsulated in a lipidic bilayer, and therefore, lipids are another important component that define exosomes. Although less studied than the RNA and protein content, research has identified several lipids that are enriched in exosome membranes. As well as reflecting the cell from which exosomes are released [31], there is an enrichment in certain lipids, such as phosphatidylserine and phosphatidylethanolamide, ceramides, or sphingomyelins [32,33].

### 2.3. Function of Exosomes

Through the transfer of cellular content between cells, exosomes are involved in the cellular communication across different cell types and different locations. Although great interest has been focused on exosomes under pathological conditions, especially cancer, these small extracellular vesicles have been identified in all cell types and in different biological states [5]. Therefore, it is not surprising that exosomes are essential mediators of cellular behaviour under normal physiological conditions. For instance, they play important roles in immune response and surveillance [34], neurotransmitter release by neurons [35], or tissue repair [36]. Similarly, cells use exosomes under pathological conditions, such as virus infections and spread [37] or cancer [7]. Furthermore, from a clinical point of view, exosomes and their cargo could be exploited as circulating biomarkers due to the ability to correlate their profile to the cell of origin [38].

Different mechanisms associated with tumorigenesis have been linked to exosome release and uptake. Amongst these, we can find extracellular remodelling, transfer of oncogenes and oncoproteins, release of pro-inflammatory factors, or even education of stroma cells into forming a pre-metastatic niche by changing their phenotype [7].

## 3. Exosomes and Metastasis

With the “seed and soil” theory from 1889, Stephen Paget hypothesised that cancer cells (seeds) were in communication with the host environment (soil), interacting to allow cancer progression [39]. Since then, extensive research in cancer has provided robust data on the interaction between cancer cells and the tumour microenvironment (TME) [40]. The TME is a complex cellular environment formed by a wide variety of cells, including endothelial cells, fibroblasts, and macrophages. It has an important role in the remodelling of cancer cells, being able to modulate their behaviour and regulate tumour progression and metastatic spread. Furthermore, different factors are released from cancer cells and stromal cells which can lead to modifications in distant sites, resulting in a favourable scenario for the formation of a pre-metastatic niche [41,42]. Amongst the factors important in this modulation of tumour and stromal cells towards the induction of metastasis and modulation of a oncogenic phenotype, exosomes have been identified (Figure 2) [43,44,45].

### 3.1. Organostropism

Exosomes can modulate the pre-metastatic niche formation via release and uptake of the exosomal cargo. In this line, different groups have evaluated if these extracellular vesicles can induce specific organ-directed metastasis (metastatic organotropism). This was first evaluated in melanoma-derived exosomes. Injection in mice resulted in homing of melanoma exosomes to sentinel lymph nodes, in comparison to liposomes which did not accumulate in these nodes [46]. Moreover, uptake of exosomes in lymph nodes resulted in extracellular matrix (ECM) degradation, vasculature growth, and signalling modifications [46], suggesting that exosomes help create a favourable setting for metastatic spread. Similar results were seen in pancreatic ductal adenocarcinoma (PDA). Selective uptake of PDA-derived exosomes by liver Kupffer cells leads to the activation of different signalling pathways resulting in fibronectin accumulation in hepatic stellate cells, increasing the recruitment of bone marrow (BM)-derived macrophages, and a subsequent increase in liver metastasis [47]. The latter was induced by the transfer of migration inhibitory factor to liver cells, as blocking of this factor inhibited metastasis in vivo [47]. In gastric cancer (GC), EGFR-containing exosomes from GC fused with liver stromal cells resulted in the transfer of EGFR to these cell’s membrane, inducing the suppression of miR-26a/b expression [48]. This induced the activation of hepatocyte growth factor and subsequent enhancement of liver metastasis by GC cells [48]. In cervical squamous cell carcinoma (SCC), miR-221-3p was enriched in these cells and transferred to lymphatic endothelial cells, promoting migration and angiogenesis in vitro, as well as the induction of lymph node metastasis in vivo [49].

Based on the published studies in organotropism, all evidence pointed to the role exosomes have in inducing metastasis in specific organs, but no data indicated what was driving the specific organotropism. This changed with the discovery by Hoshino et al. [50]. They identified that integrins present on the surface of cancer-derived exosomes were important for the destination of metastatic cells. Using exosomes from different cancer cell lines that preferably metastasised to the lung, liver, or brain, they studied the biodistribution of these exosomes and their role in the process of metastasis when injected in mice. Exosomes from the different cell lines homed to the organ site for which each cancer preferably metastasised. Moreover, this uptake took place in a specific subset of cells in each organ: lung fibroblasts and epithelial cells in the lung; liver Kupffer cells in the liver; and brain endothelial cells in the brain. The uptake of these integrin-specific exosome induced the upregulation of genes associated with migration and inflammation [50]. A proteomic analysis revealed a differential integrin expression pattern on the surface of exosomes depending on the host organ: ITGα_6_β_4_ and ITGα_6_β_1_ directed exosomes to lung metastasis, ITGβ_3_ to brain metastasis, and ITGα_ν_β_5_ to liver metastasis. Inhibition of these integrins resulted in a reduction of the exosome uptake and decrease of metastasis in mice [50]. Other exosome-associated integrins have been linked to organotropism, such as ITGα_4_ for lymph nodes, ITGβ_1_ for the brain, and ITGα_4_β_1_ and ITGα_ν_β_3_ for bone metastasis [51].

### 3.2. Neo-Angiogenesis

Angiogenesis is an important hallmark for cancer development, as the creation of new blood vessels increases the availability of nutrient and oxygen supply to the growing cellular mass [41]. This process is tightly associated with hypoxia (low oxygen supply), which can lead to changes in gene expression and phenotypic modifications [52]. In this direction, different studies have identified the importance of cancer-derived exosomes in regulating the neoangiogenic process helping tumour cells to invade other areas and increase the nutrient supply.

The uptake of exosomal miRNAs has been associated with induction of vascular permeability and angiogenesis linked to metastasis. Endothelial cells take up breast cancer (BC)-derived exosomes leading to an increase in migration, driven by the incorporation of miR-105, associated with migration and tight junction protein ZO-1 [53]. Consequently, the destruction of tight junctions and subsequent alteration of the endothelial barriers lead to an increase in vascular permeability and metastasis in vivo [53]. Similar results were seen in colorectal cancer (CRC), in which exosomal miR-25-3p was taken up by endothelial cells resulting in a pre-metastatic niche formation [54]. This miRNA has previously been associated with the regulation of vascular-associated proteins (i.e., ZO-1 or VEGFR2). Therefore, the uptake of CRC-derived exosomes containing miR-25-3p led to the disruption of vascular endothelial barriers and an increase in angiogenesis [54]. When injected in mice, CRC exosomes induced an increase in metastasis compared to exosome controls [54]. The induction of angiogenesis via exosomal transfer has also been observed in lung adenocarcinoma (LAC), where miR-142-3p is transferred to endothelial cells inducing angiogenesis [55]. LAC-derived exosomes taken up by lung fibroblasts resulted in the transformation of these cells towards a cancer-associated fibroblast (CAF) phenotype [55]. In rhabdomyosarcoma, the transfer of the miRNA cargo to fibroblasts induced an increase in proliferation, migration, invasion, and angiogenesis in vitro compared to exposition to exosome-free media [56].

Other nucleic acids than miRNAs have been associated with exosome modulation of recipient cells. Endothelial cells can also be modified by the uptake of epithelial ovarian cancer-derived exosomes via the transfer of MALAT1 lncRNA [57]. This resulted in an increase in angiogenesis in vitro and an increase in tumour growth and the density of blood vessels in vivo compared to controls [57]. In glioma, the transfer of linc-CCAT2 to endothelial cells via glioma-derived exosomes resulted in the promotion of angiogenesis via the upregulation of VEGF, TGFβ, and Bcl-2 as well as a reduction of apoptosis via Bax and caspase-3 in vitro [58].

As mentioned earlier, hypoxia plays an important role in angiogenesis induction and metastasis [52]. Different groups have investigated the role hypoxia could have in exosome transfer and the induction of change upon uptake by other cells. Work in hypoxic breast [59], lung [60], and colorectal [61] cancer-derived exosomes has identified different miRNAs to play a role in angiogenesis. Exosomes from hypoxic BC cells released miR-201 once taken up by TME cells, inducing increased expression of genes involved in the vasculature remodelling, as well as an increased proliferation, migration, and induction of capillary-like structures in vitro compared to cells not exposed to exosomes [59]. Similarly, exosomes from hypoxic lung cancer cells had increased levels of miR-23a compared to normoxic conditions [60]. When miR-23a was transferred to endothelial cells via exosomes, this induced an accumulation of HIF1α in recipient cells, triggering a signalling cascade that resulted in an angiogenic response in endothelial cells and the destruction of thigh junctions via inhibition of ZO-1 [60]. To confirm the role of this miR-23a in the angiogenic process, the inhibition of miR-23a in mice models reduced growth and angiogenesis in vivo [60]. Exosomes from hypoxic CRC cells induced migration and invasion when taken up by CRC under normoxic conditions via the Wnt4-HIF1α signalling cascade [61].

### 3.3. Induction of Pro-Metastatic Phenotypes

Exosomes contain a variety of RNAs and proteins in their cargo and present on their surface membrane [9]. Therefore, the release or interaction of such molecules with recipient cells could lead to changes in the cellular behaviour, resulting in more aggressive phenotypes [5].

Transfer of metastatic BC-derived exosomes containing Caveolin-1 to BC cells without Caveolin-1 expression resulted in increased migration and invasion, which the authors linked to several adhesion-associated proteins contained in these exosomes [62]. In another study on BC, the transfer of miR-200 from metastatic to non-metastatic cells via exosomes resulted in alterations in the gene expression profile of recipient cells, followed by an induction of mesenchymal-to-epithelial transition (MET) [63]. This transfer conferred an enhanced ability to metastasise and colonise other areas in mice models. In melanoma, cells expressing Rab27A produced exosomes packed with several proteins associated with a pro-invasive and motility phenotype, which upon uptake, increased the invasion capacity Rab27A-low melanoma cells [64]. This involvement of Rab27A in pro-metastatic changes was confirmed when knock-down of Rab27A in melanoma mice models resulted in decreased metastasis formation in vivo [64]. In hepatocellular carcinoma, the uptake of exosomes from adherent hepatocellular carcinoma cells by detached cancer cells resulted in the release of adhesion-associated proteins and mRNA that induced the attachment of these cells, which was in turn associated with lung metastasis in vivo [65]. In ovarian cancer, injection of exosomes from highly metastatic cell lines in mice models resulted in increased ability to metastasise when compared with exosomes from low-metastatic-capacity cells [66].

### 3.4. Stromal–Tumour Interactions

Not only do tumour-derived exosomes play a role in tumorigenesis and metastatic spread, but exosomes released from cells other than cancer cells can send messages to tumour cells, altering their cellular behaviour and even leading to more aggressive states [5].

In a study on CRC, exosomes from M2 macrophages were found to play a role in metastasis development. High levels of exosomal miR-21-5p and miR-155-5p were transferred from M2 macrophages to CRC cells, leading to an increase in migration and invasion via the direct downregulation of the BRG1 protein, which has been involved in CRC metastasis [67]. In a similar study, CRC-derived exosomes could educate macrophages to induce a pro-metastatic phenotype [68]. The uptake of miR-21-5p from CRC-derived exosomes to liver macrophages induced macrophage polarization, which in turn enhanced the secretion of IL-6, resulting in a proinflammatory phenotype. This process in the liver is essential for the development of liver metastasis through the miR21-TLR7-IL6 signalling axis [68].

Stromal fibroblast-derived exosomes from SCC patients contain high levels of TGFβRII, which in turn are transferred to SCC cells increasing the signalling of TGFβ, which is an important player in this malignancy [69]. A study on lung carcinoma and melanoma showed that exosomes derived from cancer cells could trigger the activation of TLR3 in recipient lung epithelial cells via the release of TLR3-ligand exosomal RNAs [70]. Activation of TLR3-induced neutrophil recruitment and chemokine production in the lung microenvironment resulted in the formation of a pre-metastatic niche in this organ. In melanoma, exosomes released from highly metastatic cells transferred their cargo, containing the MET oncogene, to BM-progenitor cells [71]. MET uptake led to a vascular leakiness and a pro-metastatic phenotype in pre-metastatic sites in vivo. Moreover, an enhanced induction of lung metastasis was seen when educated BM progenitor cells were injected in mice compared to control BM cells [71]. A study on senescent cells identified the transfer of EphA2 to active cancer cells, inducing a signalling cascade via EphA2/ephrin-A1 which resulted in an increased cellular proliferation in the recipient cell, thus promoting a tumorigenic transformation [72]. Another method by which cancer-derived exosomes can reprogram surrounding cells is via metabolism regulation. Through the delivery of miR-122 from BC cells to stromal cells, glucose consumption is reprogrammed via the downregulation of pyruvate kinase, leading to an increase in the metastasis in vivo [73].

## 4. The Role of Exosomes in Bone Sarcoma Metastasis

Bone sarcoma metastasis represents the most adverse clinical factor and is associated with poor survival rates [1]. This is partly due to the lack of understanding on the molecular mechanisms behind tumour dissemination and metastatic disease [1,74,75]. In both OS and ES, the most common metastatic site is the lung, followed by bone and bone marrow. Interestingly, lung metastasis is associated with better prognosis than non-lung metastasis [1,76].

Due to the importance of metastasis in these bone sarcomas, many studies have focused on deciphering the pathways behind disease progression. Amongst the different genes identified in OS, we find overexpression of *CD155* [77], loss of *TP53*, *RB1,* and *PTEN* [78], and upregulation of Notch genes [79] to be important in metastatic OS. Moreover, different novel treatment strategies are evaluating their efficacy in OS metastatic patients through clinical trials [75], with the aim of finding novel treatment strategies for these patients. In ES, similar studies have resulted in several genes linked to metastasis, such as *ROR1* [80], *MSH2*, *MSH6*, *RPA2,* and *RFC2* genes from the mismatch repair pathway [81], PPP1R1A [82] and TWIST1 proteins [83], the Cad11 adhesion molecule [84], or ERBB4 via activation of the PI3K-Akt-FAK cascade [85]. Moreover, similar to other cancers, hypoxia has been associated with induction of metastasis in OS and ES via regulation of HIF1α through HIF1α [86] or overexpression of CXCR4 in ES [87], amongst others. Similar to OS, different clinical trials are evaluating the response of metastatic ES patients to different treatment approaches in order to improve survival rates [76]. In contrast to OS and ES, CS is usually a non-metastatic disease with locally aggressive tumours [88]. However, some evidence suggests integrins are involved in the metastatic organotropism of CS to the lungs [89,90].

### 4.1. Exosomes in OS

OS is the most common primary bone tumour, with an incidence of 0.3 per 100.000 people per year [91]. The peak incidence follows a bimodal distribution, with most cases between 0–24 years of age and 60–85 years of age [92]. It is an intraosseous neoplasm with origin in bone regions with active cellular growth in which the balance between osteoclasts and osteoblasts is disrupted [93]. Contrary to other sarcomas with a clear genetic driving event, OS is characterised by high genomic instability. This results in complex karyotypes involving copy number alterations [94] and frequency of chromotripsis (high incidence of chromosomal rearrangements in a delimited genomic region) [95]. All these factors, adding to the heterogeneity and rarity of OS, have made difficult the successful identification of better treatment strategies for patients.

Despite improvements in diagnosis and treatment over the last decades, survival rates are still poor for an important fraction of patients. Between 10% and 20% of patients have metastasis at diagnosis, with 5-year overall survival rates (OSR) lower than 20% [75]. For localised disease at diagnosis, survival rates are higher (5-year OSR: 60–80%), although evidence suggests 80% of patients have micrometastasis at the time of diagnosis which will be refractory to chemotherapy. This results in 30–40% of non-metastatic OS patients developing metastasis and recurrent disease [96,97]. Therefore, one of the focal points of OS research is to better understand the process of metastasis and how different factors modulate the TME to favour metastatic spread. This will help in identifying treatment strategies against metastatic and refractory disease, leading to improved survival rates.

Due to the role exosomes play in tumorigenesis in many cancers and the implications these small extracellular vesicles have in metastasis, it is not strange that researchers in the OS field are investigating the implications of exosomes in OS (Table 1). As exosomes are involved in cellular cross-talk via transfer of their cargo, one of the focal points has been to characterise the cargo of OS-derived exosomes. An analysis of OS-derived exosomes and conditioned media (exosome-free) revealed a different protein profile between both [98]. Among the 250 most enriched proteins in exosomes, pathway analysis revealed an association with migration, adhesion, and angiogenesis, all important processes for cancer dissemination and modulation of the pre-metastatic niche. Following these data, the same group investigated the exosomal cargo of several OS cell lines [99]. A difference in the miRNA content was seen between metastatic and non-metastatic OS-derived exosomes. Amongst the predicted target genes for the differential expressed miRNA, there was an enrichment of genes associated with tumour progression and metastasis. A deeper analysis of miRNAs enriched in exosomes from the most metastatic OS cell line (SAOS2) revealed that 4 miRNAs targeted 31 target genes from the same network associated with cellular adhesion and apoptosis [99]. Therefore, data from these studies suggest that OS-derived exosomes could drive a pro-metastatic phenotype by transferring specific proteins and miRNA to other OS cells, resulting in induction of changes in migration, adhesion, and angiogenesis.

Similar to what has been described in other cancers, tumour-derived exosomes can educate the TME in order to induce pro-metastatic and tumorigenic changes [52]. OS-derived exosomes from metastatic cells induced an increase in migration and invasion capacity of recipient osteoblasts compared to the uptake of non-metastatic OS-derived exosomes [100]. In agreement with previous data [99], the miRNA profile of exosomes from metastatic and non-metastatic OS cells differed, with an enrichment of target genes associated with metastasis and cancer. Amongst these, miR-675 was highly present in metastatic OS-derived exosomes. Validation studies using metastatic and non-metastatic exosome transfer and mimic treatments confirmed that the uptake of miR-675 via exosomes led to the downregulation of CALN1 expression associated with migration [100]. A more recent study has investigated if OS-derived exosomes could induce changes to macrophages, osteoclasts, and endothelial cells, important components of the TME [101]. Exosome internalization by macrophages induced osteoclast gene expression profile and an increase in the number of osteoclast-like cells. In addition, osteoclasts cultured with OS-derived exosomes had higher bone resorption activity in vitro [101], suggesting that OS could induce changes in the bone microenvironment via macrophage and osteoclast modulation. Moreover, the uptake of OS-derived exosomes by endothelial cells induced an increase in pro-angiogenic factors and the formation of capillary structures in vitro, thus confirming that OS could modulate the invasiveness of cancer cells by modulating the TME [101]. Profiling of the miRNA content of these exosomes identified miR-148a and miR-21-5p as being behind the change in the phenotype of osteoclast and endothelial activities upon exosome uptake, as overexpression of these miRNAs in recipient cells induced the expression of osteoclast markers and increased bone resorption and angiogenesis in vitro [101]. Another example of OS-derived exosomes educating stromal cells was seen in mesenchymal stem cells (MSCs). Injection of MSCs co-cultured with metastatic OS-derived exosomes in an OS mouse model induced an increase in the tumour growth and tumour volume in vivo as well as metastatic dissemination to the lungs [102]. Exosome profiling identified TGFβ to be present in the exosome surface, which, once in contact with MSCs, induced a signalling cascade resulting in the increase of cytokine production (IL-6 and IL-8) by MSCs [102]. As OS cells do not produce IL-6, this release from MSCs could enhance a pro-inflammatory microenvironment favourable for metastasis.

Although not focused on the induction of a pro-metastatic phenotype in OS, Torregiani et al. [103] identified that the transfer of exosomes from doxorubicin-resistant to chemosensitive OS cells induced an increase in the resistance to chemotherapeutics in vitro. The presence of MDR-1 and Pgp mRNA, both associated with drug resistance, was confirmed in the exosomal cargo, leading to an increase in the expression of these in recipient OS-chemosensitive cells [103]. Therefore, exosomes from OS cells can not only educate TME cells, inducing a pro-metastatic and tumorigenic phenotype, but can educate less aggressive OS cells towards a chemo-resistant profile.

Another possibility is for exosomes released from TME-forming cells to modulate the behaviour of OS cells. In this direction, two different studies on OS have elucidated how non-OS exosomes can be taken up by OS cells inducing a pro-metastatic profile. In this line, Endo-Munoz et al. identified the urokinase plasminogen activator (uPA) and its receptor (uPAR) to be behind the metastatic potential of OS cells [104]. Metastatic OS cells as well as exosomes derived from these contain high levels of uPA, which have been associated with a signalling cascade inducing an increase in the migration potential in vitro and metastasis in vivo [104]. A paracrine loop in uPA signalling was identified via BM-educated OS cells. uPA was secreted by BM cells and internalised by OS cells, inducing an increase in the migration capacity and metastasis [104]. This ability of stromal cells to induce changes in OS cells has been reported in CAF-secreted exosomes. Transfer of the exosomal cargo from CAF to OS cells increased migration and invasion in vitro compared to incubation with non-cancer fibroblast [105]. This was induced by an enrichment of miR-1228 in CAF, which was transferred to OS cells via exosomes, resulting in a decrease in suppressor of cancer cell invasion expression and an increase in the migration capacity [105]. This was validated by the blocking of the migration capacity when OS cells were incubated with normal fibroblasts (not enriched by miR-1228) or when expression of miR-1228 was inhibited [105]. All these data further confirm that stromal cells can educate OS cells towards a more oncogenic phenotype, thus evidencing the importance of not only studying cancer cells but also stromal cells.

Despite not being associated with the metastatic potential of exosomes in OS, three other studies have investigated OS-derived exosomes. Shimbo et al. evaluated the ability to introduce synthetic miRNA to OS cells via exosomes in order to reduce their tumorigenic potential [106]. Inducing the expression of miR-143 in MSCs, this miRNA was transferred to OS cells, with a subsequent reduction of the migration capacity without affecting the proliferation rate in vitro. Although the delivery efficiency was lower than using lipofection, the reduction in migration was similar and with less cytotoxicity [106]. The two other studies investigated the ability to use OS-derived exosomes as non-invasive circulating biomarkers for disease progression [107] and patient stratification between chemosensitive and chemoresistant profiles [108]. This could eventually help in disease monitoring and treatment stratification in OS patients. A summary of all the data on exosomes in OS is presented in Table 1.

### 4.2. Exosomes in ES

ES is the second most common type of bone tumour in children and young adults, with its peak incidence between 15 and 24 years of age [3]. The main driving event is the translocation between the *EWSR1* gene in chromosome 22 and a member of the ETS family of transcription factors, being in 85% of cases the *FLI1* gene located on chromosome 11 [109]. However, less common ETS members have been identified as well, such as *ERG*, *ETV1*, *E1AF,* or *FEV* [110]. This results in an aberrant transcription factor fundamental for ES tumorigenic transformation [74]. Moreover, the cell-surface glycoprotein CD99 is expressed at high levels in 95–100% of cases [111] and has been involved ES tumorigenesis [112,113]. These have become two powerful diagnostic biomarkers to distinguish ES from other small round cell sarcomas.

Although there have been improvements in the diagnosis and treatment for patients with ES, the survival is still poor [76]. Most ES patients (75%) are diagnosed with localised disease, with a 5-year OSR of 75% [114]. However, 25% of patients have metastatic disease at diagnosis, which is associated with dismissal survival rates (5-year OSR <30%) [76]. Moreover, there is a proportion of patients that have refractory disease (will not respond to treatment) or will eventually recur, both being associated with poor survival rates (5-year OSR <25%) [115]. Therefore, metastatic and recurrent disease are a main concern in ES, with researchers and clinicians studying the metastatic process to better understand its progression and to identify novel therapeutic targets. All this could ultimately lead to improved survival rates.

In parallel with the OS field, researchers in the ES field are trying to decipher the role of ES-derived exosomes in tumour progression and metastasis. Although there are fewer studies than on OS, increasing evidence is suggesting an important role of these small extracellular vesicles in ES cross-talk and TME modulation. The first evidence of exosomes secreted from ES cells was in 2013 [116]. This first mention of ES-derived exosomes was centred on characterising the isolated vesicles focusing on the RNA content. This was used to define a panel of biomarkers to distinguish ES-exosomes from other cancer or non-cancer associated exosomes that could be identified in circulation [116]. Although there was no validation of the role exosomes had in ES progression, different mRNAs associated with ES tumorigenesis were identified in their cargo, such as *EWSR1-FLI1* fusion or *EZH2* [116]. Not long after this publication, another group identified *EWSR1-FLI1* mRNA to be contained in ES-derived exosomes [117]. Moreover, xenograft ES mice released ES-derived exosomes in circulation at detectable concentrations, further evidencing the possibility of using ES-derived exosomes for diagnosis and monitoring purposes. The most important finding, however, was that ES-derived exosomes could transfer the *EWSR1-FLI1* mRNA to other ES recipient cells, but not to OS cells [117]. This confirmed that exosomes could be implicated in the cross-talk between ES cells, modulating ES cellular behaviour towards a more tumorigenic state. A study using 3D tissue-engineered models based on scaffolds [118] confirmed the previous findings in monolayer cultured cell lines [116,117]. After confirming that scaffold-derived exosomes resembled more the exosomes isolated from the patient’s plasma than those from monolayer cell cultures, they confirmed that *EZH2* was present in ES-derived exosomes [118]. In contrast to previous data, ES-derived exosomes could transfer *EZH2* as part of their cargo to non-ES cells (osteoblasts and osteoclasts in this case), as well as ES cells [118]. This was the first evidence that ES-derived exosomes could transfer genetic material to non-tumorigenic cells, setting the scene for further research on the cross-talk and implications of this transfer of information.

The first study that investigated the implications of the exosomal transfer in ES was in 2015 [119]. The investigation was based in the neural differentiation triggered in ES cells by CD99 silencing via miR-34a-Notch-NFκB, antagonising the effect EWSR1-FLI1 had on ES cells [119]. However, exosomes released from CD99-silenced ES cells, once taken up by normal ES cells, could induce a change in the phenotype similar to the stable silencing of CD99. This comprised an increase in neural differentiation and a decrease in the Notch and NFκB signalling cascade, which was associated with the transfer of high levels of miR-34a via exosomes [119]. Further research from the same group showed that the uptake of CD99-silenced exosomes by ES cells induced a decrease in cellular growth, proliferation, and migration, as well as an increase in neural differentiation [120]. In contrast, when ES-derived exosomes (expressing CD99) were taken up by CD99-silenced ES cells, migration capacity was regained and a decrease in neural markers took place. As CD99 expression did not change in recipient cells after exosome uptake, this modification of the cellular behaviour had to be associated with the exosomal cargo independent of CD99, as shown before [120]. RNA and miRNA transcriptomic analysis showed that the uptake of CD99-silenced exosomes induces a change in the gene expression profile, confirming that the phenotypic alterations were driven by the transfer of genetic material [120]. Moreover, the transfer of miR-199a-3p, the most enriched miRNA in CD99-silenced exosomes, induced a similar phenotype to the uptake of the entire exosomal cargo, thus suggesting that the change in cellular behaviour was mainly induced by the transfer of this miRNA [120]. To put these data in a clinical context, the authors confirmed that tumours from patients with localised disease had higher levels of miR-199a-3p than those with metastatic disease, further confirming the role of this miRNA as a tumour suppressor. Data on the different studies on ES-derived exosomes are summarised in Table 2.

Therefore, the role that ES exosomes could play in modelling other cells and changing their cellular behaviour was confirmed. In this case, however, the change in phenotype was towards a less tumorigenic effect, which could be exploited for novel treatment strategies. Moreover, the different data on the protein and RNA content of ES-derived exosomes, including several important factors for ES tumorigenesis, suggest that this transfer of material could have important implications in TME modulation and education of neighbouring and distant cells, thus promoting the formation of pre-metastatic niches and modulation towards a more tumorigenic state.

### 4.3. Exosomes in CS

CS is the second most common type of bone sarcoma, with its peak incidence between 50–70 years of age [88]. It is a heterogeneous disease characterised by slow cellular growth, the formation of a hyaline cartilaginous tissue around the cells, and low vasculature, making this bone sarcoma highly resistant to radiotherapy and chemotherapy [88]. Moreover, 5–10% of CS patients present with advanced disease (3rd grade) associated with an elevated probability to develop metastasis [121].

In contrast to OS and ES, there are still no data on the role exosomes could play in this malignancy. However, different studies have evaluated the role of miRNAs in angiogenesis and metastatic progression in CS. Angiogenesis is associated with worse prognosis in CS [122]. It has been shown that CCL5 induces downregulation of miR-199a and miR-200b, not being able to regulate VEGF-A, resulting in higher VEGF-A levels in cells [123,124]. Similarly, miR-181a is enriched in hypoxic CS cells, which results in increased VEGF levels via the RGD16 and CXCR4 signalling cascade [125].

Not only have miRNAs been associated with angiogenesis in CS, but several studies have linked miRNAs to tumour-suppressing mechanisms. For instance, high levels of miR-129-5p induced a decrease in SOX4, resulting in the inhibition of the proliferation, migration, and induction of apoptosis [126]. A similar response was seen for miR-30a via SOX4, resulting in the induction of a similar phenotype transformation [127]. Moreover, miR-141 and miR-101 have been shown to inhibit the metastatic potential of CS via regulation of c-Src [128,129].

Based on these data and on the evidence of the role exosomes have in OS and ES, it would not be a surprise if exosomes are shown to play a role in CS tumorigenesis. The transfer of miRNAs to modulate TME or CS cells towards a pro-angiogenic state or the education of CS cells in inhibiting tumour progression by the transfer of tumour suppressor miRNAs could be identified in the future. This might help to better understand the high radiotherapeutic and chemotherapeutic resistance of CS patients to conventional treatment. Moreover, based on data that OS-derived exosomes [102] and ES-derived exosomes [119,120] can modulate cancer cells towards a less tumorigenic phenotype, reducing migration and proliferation, a similar strategy could be exploited in CS to obtain sensitivity to treatment strategies and a reduction of tumorigenesis.

### 4.4. Other Roles of Exosomes in Bone Sarcomas

As seen in other cancers, experimental evidence shows that exosomes could play an important role in disease progression and metastasis in bone sarcomas. However, the exosome biogenesis and uptake process, as well as the composition of the exosome cargo, makes these vesicles interesting candidates for bone sarcoma studies. Although it is not in the scope of this review, it is worth mentioning that exosomes derived from bone sarcoma cells are being investigated for their application as circulating biomarkers and for therapeutic strategies. Different groups are identifying specific cargo contained in ES and OS-derived exosomes that could be exploited as circulating biomarkers, thus using exosomes contained in the blood for diagnosis, disease monitoring, and patient stratification towards more personalised medicine [107,108,116,117]. Another approach that is being considered in the bone sarcoma field, but is still in its initial stages, is the use of exosomes for therapeutic strategies. Interestingly, this is being evaluated from two different perspectives. One is to exploit the innate ability of exosomes to enter specific cell populations. Using this phenomenon, researchers could load exosomes with therapeutic compounds, thus resulting in specific drug delivery and less off-target toxicity [106,120]. The other approach, although not yet investigated in the bone sarcoma field, would be the blocking of pathways important for the production or uptake of exosomes. As discussed in this review, exosomes play important roles in disease progression and modulation of the pre-metastatic niche. Therefore, targeting their production and uptake from bone sarcoma cells and the microenvironment could halt the process of metastasis [130,131].

## 5. Concluding Remarks

Metastasis is an important burden for patients in oncology. Despite improvements in diagnosis and treatment strategies over the last decades, a high proportion of cancer patients have low chances of surviving the disease. This is mainly due to the presence of metastasis disease or tumours not responding to treatments given. This is not an exception for bone sarcoma. Different directions are being considered in order to fully understand the process of metastasis and how this could be targeted, thus resulting in better survival rates. One of the biological factors that interests researchers of oncology are exosomes, as these extracellular vesicles can transfer RNAs and proteins between different cell types across the body, inducing changes in the cellular behaviour. This process has been associated with disease progression and the induction of pre-metastatic niches in many cancers, bone sarcomas amongst them. In this review, the results obtained through different studies based on exosomes in OS and ES, as well as preliminary evidence of the putative role exosomes could have in CS, have been summarised. Research in these fields has shown that exosomal cargo is enriched for proteins and RNAs involved in important processes in tumorigenesis and metastasis, such as migration or modulation of the pre-metastatic niche. Moreover, TME cells can induce changes in bone sarcoma cells, resulting in increased metastatic potential.

In our opinion, during the following years, exosome research will reveal important molecular mechanisms in cancer progression, specifically in metastasis. This will have a direct impact on bone sarcoma research, as evidence in the field is already showing that bone sarcoma-derived exosomes can modulate the TME and lead to pro-metastatic phenotypes. Based on the different findings summarised in this review, future research should focus on further deciphering how exosomes released by ES and OS cells induce a pro-metastatic phenotype, in order to pinpoint the exact molecular processes that result in disease dissemination. This could help understand the process of metastasis in bone sarcomas, providing a list of molecular targets to consider for treatment strategies. Moreover, characterising the cargo of these exosomes, in contrast to other exosome populations, we might be able to define a molecular profile uniquely associated with these sarcomas. This could be implemented as a minimally invasive strategy for diagnosis, patient stratification, and disease monitoring. Therefore, it might ultimately result in better patient follow-up and survival rates for bone sarcoma patients. Regarding the use of exosomes for therapeutic delivery or blocking exosome-associated processes to reduce tumour progression, the authors believe there is still too much uncertainty to translate the current knowledge to clinical practice. As exosomes are important in fundamental cellular processes, further research has a long way to go in terms of evaluating how to specifically target a certain subpopulation of cells or what effects blocking exosome biogenesis and uptake will have in the general cellular population.

## Figures and Tables

**Figure 1 cells-09-00241-f001:**
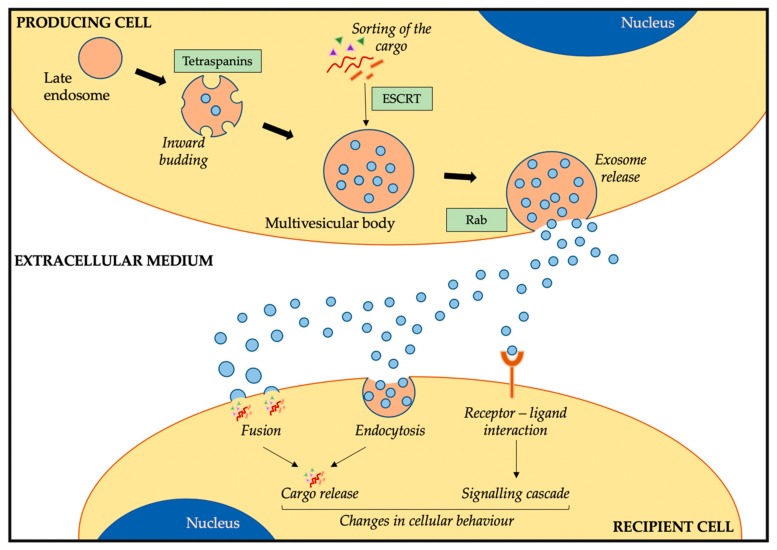
Exosome biogenesis, release, and uptake by cells. Abbreviations: ESCRT = endosomal sorting complex required for transport.

**Figure 2 cells-09-00241-f002:**
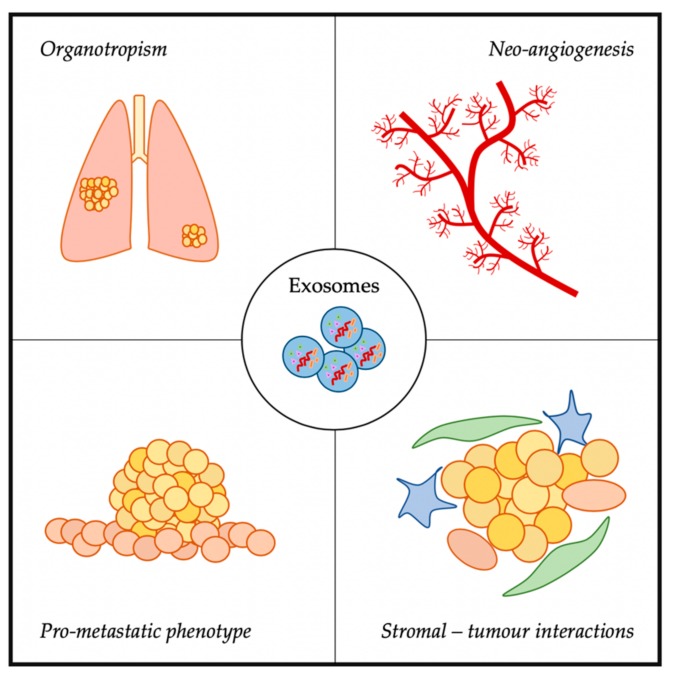
Different roles of exosomes in metastasis described in this review.

**Table 1 cells-09-00241-t001:** Summary of exosome studies in OS.

Origin Cell	Recipient Cell	Cargo	Change	Ref.
OS cells and conditioned media	-	Profiling of proteome and secretome	Exosome proteins involved in migration, adhesion, and angiogenesis	[98]
Metastatic and non-metastatic OS cell lines	-	Profiling of miRNAs and target genes	miRNA of metastatic OS exosomes target metastasis-associated genes, cell adhesion, and apoptosis	[99]
Metastatic and non-metastatic OS cell lines	Osteoblasts	miR-675 (miRNA profiling)	Metastasis-associated exosomes induce migration and invasion in osteoblasts via miR-675/CALN1 axis	[100]
Metastatic OS cell lines	Macrophages, osteoclasts, endothelial cells	miR-148a, miR-21-5p (RNA profiling)	Induction of osteoclast-like gene expression (macrophage), increase in bone resorption (osteoclasts) and angiogenesis (endothelial cells) via miRNA transfer	[101]
Metastatic and non-metastatic OS cell lines	MSC	TGFβ (induction of IL6)	Internalization of TGFβ induces IL6 production, cell growth, and lung metastasis in vivo	[102]
Doxorubicin-resistant OS cell lines	Sensitive OS cell lines	MRP1, Pgp(multidrug resistant proteins)	Increase in doxorubicin resistance in recipient cells; increase in MRP1 and Pgp mRNA levels.	[103]
Bone marrow (conditioned media)	Metastatic and non-metastatic OS cells	uPA (secreted, paracrine loop)	Increase in migration on recipient cells, induction of OS metastasis in vivo	[104]
CAF	OS cell lines	miR-1228 (miRNA profiling)	Increase in migration and invasion via miR-1228 transfer	[105]
MSC	OS cell lines	miR-143 (synthetic introduction)	Reduction of migration via exosome transfer (better than transfection)	[106]
OS cell lines	-	Profiling of miRNA as OS biomarkers	Better biomarker than ALP or patient stratification according to chemotherapy response	[107,108],

Abbreviations: Ref = reference, MSC = mesenchymal stem cells, CAF = cancer-associated fibroblasts.

**Table 2 cells-09-00241-t002:** Summary of exosome studies on ES.

Origin cell	Recipient cell	Cargo	Change	Ref.
ES cell lines	-	EWSR1-FLI1, EZH2, and 10 more mRNAs (mRNA profiling of known ES targets)	Suitable as circulating biomarkers for ES, detectable in spike-in healthy blood samples	[116]
ES cell lines	ES cell lines	EWSR1-FLI1 mRNA	Labelled EWSR1-FLI1 transferred to other ES cells, not to OS	[117]
ES cell lines	Osteoblasts, osteoclasts in 3D scaffold	EZH2 mRNA (target of interest)	Transfer of EZH2 mRNA to MSC (increase expression), osteoblasts (no change), and osteoclasts (reduction expression)	[118]
ES^CD99neg^ cell line model	ES cell lines (normal CD99)	Increased miR-34a	Regulation of NFκB via miR-34a through reduction of Notch. Increase in neural differentiation (similar to direct CD99 silencing)	[119]
ES^CD99neg^ cell line model	ES cell lines (normal CD99)	miR-199a-3p (miRNA profiling) miR-199a-3p	Induction of different gene expression profiles, neural differentiation, and neurogenesis; reduction of cell growth and migration (similar to miR mimic)	[120]

Abbreviations: Ref = reference, neg = negative.

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
