# Peer review of "Exosomes in Bone Sarcomas: Key Players in Metastasis"

_cells, 2020, doi:10.3390/cells9010241_

Round 1

Reviewer 1 Report

The manuscript by Chicon-Bosch and Tirado is generally a well-written, well-referenced and comprehensive review on the role of exosomes in bone sarcomas, with a particular focus on their role in sarcoma metastasis. The review will be of interest to the field.

However, a few minor corrections should be done before publication:

1) The authors should stick to the WHO terminology. As such, the preferred term for Ewing sarcoma is not Ewing's sarcoma, but simply Ewing sarcoma. In addition, abbreviations such as "ES" should be used consistently throughout the text.

2) The overall level of English is good. However, the manuscript could benefit from proof-reading by a native English speaker to shorten some sentences and to enhance the flow of reading.

3) Chapters 2 and 3 could be shortened as they are not focussed on bone sarcomas and since there are extensive reviews on the mentioned topics in the literature.

4) The review focusses on the role of exosomes in bone sarcoma metastasis in chapter 4. This Reviewer recommends to broaden the view of this chapter and to report also on the general role of exosomes in bone sarcomas (i.e. their roles in bone sarcomas apart from their role in metastasis). What aspects concerning exosomes in bone sarcomas might be different from other cancer types?

5) The chapter 5 "concluding remarks" is a bit generic and should include a stronger perspective from the authors on what they consider being the key next steps to define the role of exosomes in bone sarcomas in general and in particular in metastasis. What kind of experiments need to be carried out? What do the authors expect to find that might be different from other cancer entities such as carcinomas? How can these findings be implemented in clinical diagnostics and therapy?

Reviewer 2 Report

The manuscript provides a comprehensive and timely review on the role of exosomes in the oncology field, providing recent updates on the contribute of exosomes in inducing metastatic progression of bone sarcomas. This topic is timely and concerns studies relevant to clinical oncology. The literature on this subject is almost completely covered and condensed into a clear presentation.

Minor points:

Summary of exosome studies in osteosarcoma (Table 1) has been mentioned in the text on page 8 (line 302) as Table 2. Regarding the question on cellular mechanisms by which exosomes internalize, on page 3, a recent paper from Horibe S.could be mentioned. (Horibe, S., Tanahashi, T., Kawauchi, S. et al. Mechanism of recipient cell-dependent differences in exosome uptake. BMC Cancer 18, 47 (2018) doi:10.1186/s12885-017-3958-1). Regarding the tumor-stroma cross-talk which has been discussed on page 4, in my opinion   the Reference 40 should be complemented by the article:  Liu Y, Cao X. Characteristics and Significance of the Pre-metastatic Niche. Cancer Cell (2016) 30:668–681, in which the temporal sequence of the key events occurring during the metastatic niche formation are deeply discussed.

Reviewer 3 Report

This is a well-written and comprehensive review which covers the role of exosomes in bone tumors with a look for their implications in promoting metastatization. The scientific content is surely of great interest and will capture the interest of a broad audience.

Being said that, I would suggest

to add a short but detailed description  of metastasis derived from bone tumors with an additional figure which resumes what is actually known on this specific type of metastasis. This will facilitate for the general audience the understanding of the more speculative section of the review which deals with the potential role of exosomes in bone tumors metastasis; Any therapeutic speculation/implication on targeting exosomes. In such a detailed review this might be a further plus.
